# Altered Gut Microbiome Composition in Dogs with Hyperadrenocorticism: Key Bacterial Genera Analysis

**DOI:** 10.3390/ani14192883

**Published:** 2024-10-07

**Authors:** Hee-Jun Kang, Sang-Won Kim, Seon-Myung Kim, Tae-Min La, Jae-Eun Hyun, Sang-Won Lee, Jung-Hyun Kim

**Affiliations:** 1Department of Veterinary Internal Medicine, College of Veterinary Medicine, Konkuk University, Seoul 05029, Republic of Korea; tobaren4@konkuk.ac.kr (H.-J.K.); everyday54@konkuk.ac.kr (S.-W.K.); jaeeunhyun@konkuk.ac.kr (J.-E.H.); 2KR Lab Bio Incorporation, Suwon 16429, Republic of Korea; smkim@krlab.bio; 3Department of Veterinary Microbiology, College of Veterinary Medicine, Konkuk University, Seoul 05029, Republic of Korea; fkxoals@konkuk.ac.kr (T.-M.L.); odssey@konkuk.ac.kr (S.-W.L.)

**Keywords:** hyperadrenocorticism, hypercortisolism, microbiome, dysbiosis, trilostane, cortisol

## Abstract

**Simple Summary:**

Hyperadrenocorticism in dogs is a condition characterized by excess cortisol and can disrupt overall health. This study examined its impact on gut bacteria, which is crucial for health. Compared to healthy dogs, the dogs with hyperadrenocorticism had less diverse gut bacteria and an increase in Proteobacteria (Pseudomonadota), Actinobacteria, *Bacteroides*, *Enterococcus*, *Corynebacterium*, *Escherichia*, and *Proteus*, alongside a decrease in Firmicutes (Bacillota). These imbalances continued even after treatment, suggesting ongoing health risks. Understanding these gut changes can improve treatments for affected dogs.

**Abstract:**

Hyperadrenocorticism (HAC) is a common endocrine disorder in dogs, which is associated with diverse metabolic abnormalities. We hypothesized that elevated cortisol levels in dogs with HAC disrupt the gut microbiome (GM), and this disruption persists even after trilostane treatment. This study explored GM composition in dogs with HAC. We included 24 dogs, 15 with HAC and 9 healthy controls, and followed up with 5 dogs with HAC who received trilostane treatment. The GM analysis revealed significant compositional changes in dogs with HAC, including reduced microbiome diversity compared to healthy controls, particularly in rare taxa, as indicated by the Shannon index (*p* = 0.0148). Beta diversity analysis further showed a distinct clustering of microbiomes in dogs with HAC, separating them from healthy dogs (*p* < 0.003). Specifically, an overrepresentation of Proteobacteria (Pseudomonadota), Actinobacteria, *Bacteroides*, *Enterococcus*, *Corynebacterium*, *Escherichia*, and *Proteus* populations occurred alongside a decreased Firmicutes (Bacillota) population. Despite trilostane treatment, gut dysbiosis persisted in dogs with HAC at a median of 41 d post treatment, suggesting its potential role in ongoing metabolic issues. We identified GM dysbiosis in dogs with HAC by examining key bacterial genera, offering insights into potential interventions like probiotics or fecal microbiota transplants for better HAC management.

## 1. Introduction

Canine hyperadrenocorticism (HAC), characterized by excessive cortisol production leading to chronic hypercortisolism, is a significant endocrine disorder in dogs [1]. Dogs with HAC exhibit symptoms similar to those seen in human HAC, known as Cushing’s syndrome (CS). These symptoms include systemic hypertension, hyperglycemia, hypertriglyceridemia, hypercholesterolemia, insulin resistance, centripetal redistribution of body fat, hypercoagulability, and steroid hepatopathy [2].

Lifelong medical therapy can effectively manage the clinical signs of HAC in most dogs. However, it may not fully reverse the metabolic abnormalities [3,4].Systemic comorbidities often persist in humans despite successful management of CS [5].Therefore, understanding the underlying mechanisms and developing novel approaches to address glucocorticoid-induced metabolic issues is crucial. Exploring potential interventions presents promising strategies for preventing and treating metabolic problems associated with excess cortisol.

Recent advancements in human medicine have highlighted the emerging correlation between endocrine disorders and the gut microbiome (GM) [6]. The intricate interaction between the gut–brain axis and the hypothalamic–pituitary–adrenal (HPA) axis, which regulates the stress response, has become increasingly recognized [7,8,9,10]. Notably, the GM modulates stress levels, while steroid levels impact GM composition [11]. Recent studies have revealed associations between GM and metabolic abnormalities related to HAC [5,12,13]. Specific bacteria capable of degrading cortisol have been identified in patients with HAC, suggesting GM’s potential role in regulating steroid hormone levels and overall health [13]. Furthermore, long-term remission in patients with Cushing’s disease does not necessarily restore a healthy GM, which may contribute to persistent cardiometabolic risks [5]. GM imbalance and reduced propionic acid levels have been observed in CS, suggesting that GM modulation could be a promising therapy for associated metabolic issues [12].

Despite the established link between GM and HAC in humans, research on this topic in dogs is lacking. We hypothesize that elevated cortisol levels in HAC disrupt the GM, and this disruption persists even after trilostane treatment. To test this hypothesis, we analyzed the GM of dogs with HAC and compared their clinical features and GM composition with those of healthy dogs. We also examined the GM of dogs with HAC before and after trilostane treatment to explore the potential role of GM dysbiosis in sustaining these metabolic problems.

## 2. Materials and Methods

### 2.1. Animals

This prospective study was conducted at the Konkuk University Veterinary Medical Teaching Hospital between June 2023 and June 2024. The Konkuk University Institutional Animal Care and Use Committee reviewed and approved all protocols (approval number: KU23131-1). We obtained informed consent from each dog’s companion before including their pet in the study.

#### 2.1.1. Healthy Group

The healthy dogs were selected based on their medical history, physical examination, complete blood count (CBC), and serum biochemistry results. We assessed the clinical characteristics of the dogs, including age, sex, breed, fecal score, body condition score (BCS), body weight (BW), and systolic blood pressure (SBP), to ensure they met the study’s inclusion criteria. We excluded dogs that were pregnant, diagnosed with symptomatic urinary tract infections, cancer, hormonal disorders, kidney disease, and gastrointestinal conditions, or had received medical treatment affecting intestinal function within the past 4 weeks, including antibiotics or probiotics. We included dogs that were fed only prescribed or commercial diets.

#### 2.1.2. HAC Group

HAC was diagnosed in accordance with the American College of Veterinary Internal Medicine Consensus statement [14]. The inclusion criteria were: (1) typical HAC symptoms, such as polydipsia, polyuria, polyphagia, panting, abdominal distension, endocrine alopecia, hepatomegaly, muscle weakness, and systemic hypertension; (2) laboratory findings indicative of HAC, including neutrophilic leukocytosis, lymphopenia, eosinopenia, thrombocytosis, mild erythrocytosis, increased alkaline phosphatase (ALP) and alanine aminotransferase (ALT) levels, hypercholesterolemia, hypertriglyceridemia, hyperglycemia, urine specific gravity (USG) ranging from ≤1.018 to 1.020, and proteinuria; (3) lack of cortisol suppression following low-dose (0.01 mg/kg) dexamethasone administration (≥14 μg/L) or increased cortisol levels (>220 μg/L) after adrenocorticotropic hormone (ACTH) stimulation; and (4) absence of systemic or exogenous glucocorticoids. We differentiated between pituitary-dependent HAC (PDH) and functional adrenal tumor based on the ultrasonographic characteristics of the adrenal glands [15] or the outcomes of low-dose dexamethasone suppression testing (4 h cortisol < 14 μg/L or 4 and 8 h cortisol ≤ 50% of baseline cortisol) [16].

The exclusion criteria and diet for dogs with HAC were identical to those applied to the healthy group.

### 2.2. Clinical and Laboratory Assessments and Evaluation of Clinical Outcomes Following Trilostane Therapy

We performed comprehensive clinical and laboratory assessments, including a thorough physical examination, blood tests, and an ACTH stimulation test for each dog. The physical examination evaluated the overall health status of the dogs, while blood tests provided insights into potential abnormalities and renal function. Blood samples were collected from the jugular vein. CBC and biochemical analyses were performed using a ProCyte Dx Hematology Analyzer and a Catalyst One Chemistry Analyzer (both by IDEXX Laboratories Inc., Westbrook, ME, USA), respectively. Baseline cortisol concentration was measured before administering synthetic ACTH (Synacthen, Alfasigma, Bologna, Italy) at a dose of 5 μg/g intravenously. Stimulated cortisol concentration was measured after administering ACTH. Serum cortisol concentration was measured using an in-house enzyme-linked immunosorbent assay kit (IDEXX SNAPshot Dx Cortisol Test, IDEXX Laboratories, Westbrook, ME, USA). The optimal target post-ACTH cortisol concentration for dogs undergoing HAC treatment was <55 μg/L. The trilostane dose remained unchanged if reasonable clinical control was achieved, with post-ACTH cortisol concentrations reaching <90 μg/L. However, if post-ACTH cortisol levels and clinical control were unsatisfactory, the trilostane dose was increased accordingly [4].

Dogs with HAC underwent a trilostane therapy protocol with a starting dose of 1 mg/kg of BW, administered every 12 h, commencing on the day after diagnosis. Treatment response was evaluated every 2 weeks to monitor cortisol levels, and the trilostane dosage was adjusted if necessary. Adjustments were made based on the patient’s clinical response until a stable and effective dose was established. The first recheck was primarily to ensure that overdosing had not occurred. An ACTH stimulation test was conducted 4–6 h after the morning oral administration of trilostane (Vetoryl, Dechra, Northwich, UK), with the dose adjusted individually for each dog, to assess the adrenal response and measure cortisol levels. At each reevaluation, the companions participated in the assessment, providing information regarding their dog’s general well-being, changes in clinical signs, and any adverse effects of trilostane treatment. Based on the companion’s observations, we evaluated the clinical signs associated with HAC, including polydipsia, polyuria, polyphagia, and panting. The companions completed a standardized questionnaire before and after trilostane treatment. The questionnaire employed a scale ranging from 1 to 5 to assess the severity of each clinical sign (Table 1). The degree of improvement was determined using the scores obtained, with lower scores indicating a reduction in or resolution of signs and higher scores indicating no improvement or potential worsening [4].

### 2.3. Microbiome Sample Collection

Dogs with significant gastrointestinal issues were excluded through the Nestlé Purina^®^ (St. Louis, MO, USA) fecal score evaluation to isolate the influence of gastrointestinal conditions [17]. Fecal consistency and quality were assessed using a 7-point scale (1 = hard, dry stool; 7 = watery, liquid stool). We included only dogs with scores < 5, indicating minimal to moderate stool irregularities, thus excluding potential confounding factors caused by severe gastrointestinal disturbances, thereby ensuring the experiment’s accuracy and reliability.

We collected microbiome samples from healthy dogs and dogs with HAC. The samples from dogs with HAC were collected within 24 h after diagnosis. A second sample was collected after treating the dogs with trilostane twice daily at a stable dose for at least 10 d. During the treatment period, we captured the microbiome composition once the dogs had reached a stable trilostane dose [18]. The detailed rectal sample collection process is as follows: (1) The sterile culture swab applicator was inserted into the anus to a depth of 4–5 cm and rotated gently at the anal sphincter to collect the rectal swab samples; (2) the swabs were quickly placed into transport medium (REST™ NBgene-GUT, Noblebiosciences, Hwaseong, Republic of Korea) and stored at 4 °C until further analysis. We extracted deoxyribonucleic acid (DNA) within 4 weeks of sample collection.

### 2.4. Deoxyribonucleic Acid Extraction and 16S Ribosomal Ribonucleic Acid Sequencing

Following the manufacturer’s instructions, we extracted total genomic DNA using the PureLink™ Microbiome DNA Purification Kit (Invitrogen, Carlsbad, CA, USA; A29790). DNA from each sample was eluted in 50 μL of S6 Elution buffer, and its concentration and purity were evaluated using a NanoDrop ND-1000 spectrophotometer (Thermo Fisher Scientific, Waltham, MA, USA) and Qubit fluorometer (Thermo Fisher Scientific). DNA was extracted from canine stool samples and used as a template for polymerase chain reaction (PCR) amplification targeting the V3–V4 variable regions of the bacterial 16S ribosomal ribonucleic acid (rRNA) gene, utilizing barcoded primers with adaptors designed for the Ion S5™ sequencing system (Thermo Fisher Scientific), specifically the V3 forward primer (341F: CCTACGGGNGGCWGCAG) and the V4 reverse primer (805R: GACTACHVGGGTATCTAATCC). Each reaction mix contained 23 μL of Platinum PCR SuperMix High Fidelity, 1 μL of 10 μM forward primer, 1 μL of 10 μM reverse primer, and 2 μL of 2.5 ng/μL genomic DNA template. PCR thermocycling conditions were 3 min at 94 °C, followed by 25 cycles of 30 s at 94 °C, 30 s at 50 °C, and 30 s at 72 °C; then a 5 min extension at 72 °C; and a final hold at 4 °C. After amplification, the PCR samples were purified using AMPure™ XP (Beckman Coulter Life Sciences, Indianapolis, IN, USA) reagent according to the manufacturer’s instructions. The PCR amplicons were quantified using a Qubit dsDNA HS Assay Kit (Invitrogen, Q32854), and each amplicon was diluted to 100 pM post quantification. Equal volumes of diluted amplicons were collected in 1.5 mL tubes to achieve a final volume of 50 μL.

### 2.5. Bioinformatics Analysis of 16S Metagenomic Sequencing Data

16S metagenomics sequencing was performed using the Ion S5 platform. The Torrent Suite™ Software 5.18 (Thermo Fisher Scientific), installed on the Ion S5, automated the sequencing data analysis by trimming adapters and filtering low-quality reads. The mean amplicon read length ranged from 349 to 387 bp. Quality filtering was performed according to Torrent Suite™ guidelines. We excluded low-quality reads, including those with unrecognizable key signals, low signal quality, or reads trimmed to <25 bases. Q20 quality filtering was applied, retaining reads with a predicted error rate of 1% (Q20). The BAM files produced by Torrent Suite™ were analyzed using the Ion Reporter™ software (Thermo Fisher Scientific, version 5.20.6.0). The primary microbiome analysis was conducted using QIIME [19], with the Curated MicroSEQ^®^ 16S Reference Library v2013.1 and Curated Greengenes v13.5 databases [20].

Statistical analysis was performed on the Operational Taxonomic Units (OTUs) produced by the Ion Reporter™. Using R (Version 4.3.2, R foundation for Statistical Computing, Vienna, Austria), specific filtering parameters were applied to the samples, and rarefaction was performed on the filtered OTU table to achieve an even depth. Statistical analyses for alpha and beta diversity were performed using the R packages “phyloseq” and “vegan”. The “phyloseq” package was used to calculate alpha diversity indices and to handle and visualize microbiome data, generating beta diversity plots based on Bray–Curtis, weighted, and unweighted UniFrac distance metrics. The “vegan” package was utilized for beta diversity analysis through Permutational Multivariate Analysis of Variance (PERMANOVA) to evaluate differences in microbial community composition between groups.

Additional analyses were conducted using MicrobiomeAnalyst (Version 2.0; Xia Lab, McGill University, Montreal, QC, Canada). We conducted a Permutational Analysis of Multivariate Dispersions (PERMDISP) to assess the homogeneity of multivariate dispersions among groups. To explore the potential effects of different variables on GM composition, we performed a Microbiome Regression-based Kernel Association Test (MiRKAT) analysis to examine the associations between GM composition and various explanatory variables. Differential abundance analysis was conducted using DESeq2; the results were visualized using its graphical tools. A significance level of *p* < 0.05 was applied to all analyses. Raw data were archived in the National Center for Biotechnology Information Sequence Read Archive database (PRJNA1150743).

### 2.6. Statistical Analysis

Statistical analysis was performed using SPSS, version 29 (IBM Corporation, Armonk, NY, USA). Data normality was assessed using the Shapiro–Wilk test. Normally distributed data are presented as means ± standard deviation, while non-normally distributed data are presented as medians and interquartile ranges. Differences between independent groups were analyzed using the Mann–Whitney U test for non-parametric data. For paired data within groups, normally distributed data and non-parametric data were analyzed using the paired *t*-test and the Wilcoxon signed-rank test, respectively. Graphs were created using GraphPad Prism 10 (GraphPad Software, San Diego, CA, USA). A significance level of *p* < 0.05 was applied to all analyses.

## 3. Results

### 3.1. Clinical Characteristics of the Animals

This study included 24 dogs, 15 with HAC and 9 healthy controls (Figure 1). One dog in the HAC group was confirmed to have PDH, and 14 were highly suspected of having PDH. Table 2 presents the clinical characteristic data for the 24 dogs, including age, sex, breed, fecal score, BCS, BW, SBP, and hematologic and biochemical properties. The dogs with HAC had significantly increased age (*p* = 0.0481), platelet count (*p* = 0.0251), ALP serum concentrations (*p* = 0.0017), ALT (*p* = 0.0154), and glucose levels (*p* = 0.0064). No significant differences were found in sex, fecal score, BCS, BW, SBP, baseline cortisol levels, packed cell volume, serum concentrations of cholesterol, triglyceride (TG), or USG between the healthy and HAC groups. Appendix A presents the diets of healthy dogs and dogs with HAC.

Of the 15 dogs with HAC, 5 received trilostane treatment. Table 3 presents the characteristics of these five dogs with HAC before and after the treatment. In dogs with HAC, significant reductions were observed in clinical signs score (*p* = 0.035), post-ACTH cortisol levels (*p* < 0.0001), and SBP (*p* = 0.007) following trilostane treatment.

### 3.2. Gut Microbiome Analysis

The GM was analyzed in all dogs to compare its profile in the HAC and healthy groups. We obtained 7,861,186 reads from 29 samples through 16S metagenomics sequencing. After quality filtering, 4,551,337 reads met the quality criteria for inclusion in the analysis, with an average of 271,075 reads per sample (range: 120,340–533,864). Of these, 2,316,062 reads were accurately mapped to the reference database, ensuring a high confidence level in taxonomic assignment. We excluded reads with very low copy numbers (copies < 2) from further analysis due to low reliability. In addition, 8064 valid reads remained unmapped, as they could not be assigned to any reference in the database. The coverage depth captured the diversity of microbial communities sufficiently in each sample, providing robust support for alpha and beta diversity analyses.

#### 3.2.1. Alpha Diversity Analysis of Gut Microbiome

Alpha diversity, assessed using the Chao1 and Shannon indices, revealed that dogs with HAC exhibited reduced diversity compared to healthy dogs, based on the Shannon index (Mann–Whitney test, *p* = 0.0148). However, no significant difference was observed with the Chao1 index (*p* = 0.0860) (Figure 2A,B; Appendix A).

In dogs with HAC, alpha diversity before and after trilostane treatment was compared using the Chao1 and Shannon indices. No significant differences were found in alpha diversity before and after treatment (Chao1, *p* = 0.0875; Shannon, *p* = 0.8125) (Figure 2C,D; Appendix A).

#### 3.2.2. Beta Diversity Analysis of Gut Microbiome

Beta diversity analysis, utilizing principal coordinate analysis (PCoA) with Bray–Curtis, weighted, and unweighted UniFrac distances, revealed distinct clustering of microbiome samples from dogs with HAC, separating them from the healthy group (PERMANOVA, *p* = 0.0008, *p* = 0.0029, and *p* = 0.001 for Bray–Curtis, weighted, and unweighted UniFrac distances, respectively; Figure 3). The PERMDISP showed no significant differences in dispersion among groups (*p* = 0.48221), indicating that the PERMANOVA results reflect genuine differences in microbial community composition between the HAC and healthy groups, rather than variations in within-group dispersion.

In addition, beta diversity analysis was based on PERMANOVA of Bray–Curtis distances stratified by categories of age (mature adults, senior, and geriatric), diet (Hills, Royal Canin, and other commercial diets), sex, and BCS (thin, ideal, overweight), with categories referenced from [21,22]. PCoA plots visually suggest no clear separation in GM composition among various categories (age, diet, sex, BCS). Moreover, MiRKAT analysis showed no significant differences in GM composition by age (*p* = 0.32179), diet (*p* = 0.73789), or BCS (*p* = 0.61999). Similarly, PERMANOVA analysis for sex also indicated no significant differences (*p* = 0.8725; Figure 4).

Comparing the GM of the same dogs before and after trilostane treatment, PCoA plots visually suggest no clear separation (Figure 5). The PERMANOVA analysis, which showed no significant differences in GM composition (Bray–Curtis: *p* = 0.969; weighted UniFrac: *p* = 0.9028; unweighted UniFrac: *p* = 0.954;), confirmed this result.

#### 3.2.3. Gut Microbiome Taxa Analysis

At the phylum level, dogs with HAC demonstrated a significantly higher relative abundance of Proteobacteria (*p* = 0.0059601) and Actinobacteria (*p* = 0.0092322) and a significant decrease in Firmicutes (*p* = 1.642 × 10^−5^) compared to controls (Figure 6).

At the family level, dogs with HAC demonstrated a significantly higher relative abundance of *Corynebacteriaceae* (*p* = 0.001198), *Enterococcaceae* (*p* = 0.0066325), and *Enterobacteriaceae* (*p* = 0.020587), while *Erysipelotrichaceae* (*p* = 0.0013134) and *Lachnospiraceae* (*p* = 0.0075533) decreased significantly compared to healthy dogs.

At the genus level, a significantly higher relative abundance of *Corynebacterium* (*p* = 7.897 × 10^−5^), *Proteus* (*p* = 1.104 × 10^−4^), *Enterococcus* (*p* = 8.9322 × 10^−4^), *Escherichia* (*p* = 0.0030577), and *Bacteroides* (*p* = 0.0099045) was observed (Figure 7). No significant differences or trends were observed in the GM composition at the phylum or genus level before and after HAC treatment (Figure 8).

## 4. Discussion

In this study, the predominant bacterial phyla observed in healthy dogs and dogs with HAC were Bacteroidota, Firmicutes, Fusobacteria, Proteobacteria, and Actinobacteria, consistent with findings in human and veterinary medicine [23,24,25]. We demonstrated that dogs with HAC (*n* = 15) exhibit significant gut dysbiosis, characterized by lower GM abundance and diversity compared to healthy controls (*n* = 9). The HAC group exhibited an overrepresentation of Proteobacteria, Actinobacteria, *Bacteroides*, *Enterococcus*, *Corynebacterium*, *Escherichia*, and *Proteus* populations, alongside decreased Firmicutes, compared to the healthy dogs. This finding aligns with observations in humans with CS, who exhibit similar GM alterations: an abundance of Proteobacteria and Actinobacteria and a diminished presence of Firmicutes [5,12,13].

Given the significant gut dysbiosis in dogs with HAC, elevated cortisol levels could initiate this dysbiosis. Previous studies have shown that cortisol can be detected in feces following IV administration of ACTH in dogs [26]. Human studies have also demonstrated the presence of cortisol or its metabolites in feces, indicating the presence of blood cortisol in the gut [27]. Cortisol can impact the gut directly by increasing permeability, altering nutrient availability, and affecting bile acid metabolism, thus promoting dysbiosis [28,29]. The immunosuppressive effects of cortisol can weaken gut defenses, exacerbate inflammation, and promote inflammation-associated pathobionts [30]. Furthermore, cortisol can modulate nutrient absorption and metabolism, favoring specific bacteria while hindering beneficial ones, thus contributing to dysbiosis [31]. Disrupting bile acid synthesis and metabolism creates an environment that favors harmful bacteria [32]. Indirect effects through the stress axis also contribute to gut dysbiosis. Chronic stress and elevated cortisol levels activate the HPA axis and disrupt gut–brain communication, potentially exacerbating gut dysbiosis through complex feedback loops [9,28,33]. Future research should explore fecal cortisol levels in relation to gut dysbiosis to provide valuable insights into the mechanisms underlying these associations.

The observed enrichments in Proteobacteria genera, coupled with Firmicutes depletion, underscore the need for further research into how these bacterial communities contribute to HAC-associated metabolic abnormalities. Disrupted GM, observed in animals and humans, has been implicated in metabolic abnormalities; it triggers a cascade of effects leading to metabolic abnormalities through inflammation, altered nutrient metabolism, changes in short-chain fatty acid (SCFA) production, and altered bile acid metabolism [34,35].

Elevated Proteobacteria levels, such as *Escherichia* and *Proteus*, and Actinobacteria, including *Corynebacterium*, suggest that hypercortisolism may drive gut dysbiosis characterized by these bacteria. This dysbiosis may contribute to a chronic inflammatory state and metabolic complications similar to those observed in human CS [5,12,13]. overabundance of these bacteria could be a risk factor for various metabolic diseases [36,37]. Increased Actinobacteria has been linked to cardiometabolic risk in patients with CS, reinforcing the connection between hypercortisolism and GM shifts [5]. An overabundance of Proteobacteria is a risk factor for various metabolic diseases, as lipopolysaccharides (LPSs) from Gram-negative bacteria can trigger inflammatory responses and cytokine production, potentially linking the GM to the pro-inflammatory state [38]. Moreover, research has proposed a connection between elevated Proteobacteria and chronic low-grade inflammation in obese dogs through increased LPS levels [39], suggesting that specific strains might contribute to the metabolic abnormalities observed in HAC. Previous studies have highlighted that certain Proteobacteria strains induced fat accumulation, nonalcoholic fatty liver disease, and metabolic disorders in mono-colonized germ-free mice [40]. Similarly, increased Actinobacteria levels have been linked to metabolic conditions and lipid metabolism, indicating their involvement in cortisol-driven metabolic disorders [41,42,43].

The significant enrichment of *Bacteroides* and depletion of Firmicutes in dogs with HAC align with the findings in human CS, where similar shifts and lower propionic acid levels, a beneficial SCFA crucial for metabolic health, have been observed [12]. A study linked specific gut bacteria to increased branched-chain amino acids in insulin-resistant individuals, suggesting that dysbiosis can influence blood composition and contribute to insulin resistance [38]. Decreased Firmicutes and a lower fat/bone ratio increase the possibility of reduced production of beneficial SCFAs [11]. Furthermore, studies have shown that individuals experiencing stress-induced depression have elevated cortisol levels correlating with higher Bacteroidota and lower Firmicutes, reinforcing the link between cortisol and GM composition [44]. As *Bacteroides* spp. play a role in bile acid metabolism, deoxycholic acid-induced gut dysbiosis, and stress, models suggest increased *Bacteroides* might contribute to intestinal inflammation and metabolic disturbances [45,46]. Elevated liver enzyme levels in the HAC group, compared to the control group, suggest possible bile acid metabolism disorders. In addition, *Bacteroides* enterotypes are an independent risk factor for type 2 diabetes mellitus, attributed to increased LPS levels, causing decreased insulin sensitivity [47]. In this study, we observed an increase in *Bacteroides* in dogs with HAC, where insulin resistance is commonly seen due to elevated cortisol levels contributing to decreased insulin sensitivity [48,49]. The observed shifts in GM composition highlight the critical need for further research into the contributions of these distinct bacterial communities to HAC-associated metabolic abnormalities. Exploring the association between gut microbial dysfunction and cortisol-degrading bacteria in dogs with HAC is warranted to elucidate potential underlying mechanisms. In dogs with HAC, Firmicutes were generally observed to be significantly decreased, while *Enterococcus*, a genus within this phylum, showed an increase. This increase in *Enterococcus*, despite the overall reduction in Firmicutes, can be attributed to their role as proinflammatory microbes that flourish in dysbiotic and inflammatory environments, particularly under conditions of stress or hypercortisolism, where reduced microbial diversity and metabolic disturbances favor the growth of opportunistic pathogens [50,51].

In this study, trilostane effectively managed the clinical signs of HAC in dogs. However, serum ALP levels remained elevated, and no significant reductions were observed in variables other than SBP. Despite treatment, no significant alterations were observed in the GM, indicating the need for separate monitoring and treatment for dysbiosis. According to previous studies, dogs with HAC still exhibit persistent hypertension, elevated leptin and insulin levels, and incomplete resolution of proteinuria, hypercoagulability, and inflammatory markers despite trilostane treatment, which inhibits cortisol production and alleviates clinical symptoms. This could be due to trilostane’s inability to fully normalize serum cortisol levels, with concentrations still exceeding physiological levels at certain times of the day, even when administered twice daily [4,52]. This incomplete normalization of serum cortisol levels could account for the lack of changes in the GM despite trilostane administration. Studies in humans with CS have indicated that GM disruptions can persist even after long-term remission, suggesting a potential link to sustained dysmetabolism [5]. Further research is needed to establish a definitive causal link between persistent gut dysbiosis and metabolic abnormalities in HAC. Consequently, additional treatments addressing GM disturbances seem necessary despite improved clinical symptoms through trilostane administration. Interventions such as probiotics or fecal microbiota transplants (FMTs) are beneficial, and further investigation should explore these approaches to enhance overall therapeutic outcomes. Human studies have shown that probiotic interventions, particularly with *Lactobacillus helveticus R0052* and *Bifidobacterium longum*, can reduce urinary-free cortisol levels in healthy volunteers [53]. The potential effects of FMT have been described in humans in the context of the gut–brain axis [7]. In a study by Vrieze et al. [54], significant improvements in insulin sensitivity were observed in patients with metabolic syndrome after FMT. In addition, studies in dogs suggest that altering the GM using *Lactobacillus* fermentation or gallnut tannic acid can influence the bacterial composition and cortisol levels, indicating a possible link between GM composition and cortisol levels [55,56]. This exploration could provide valuable insights into novel treatment approaches for managing HAC and associated metabolic abnormalities.

This study had some limitations. First, the sample size comprised a relatively small population of dogs with HAC, particularly for analyses comparing pre- and post-treatment states. Second, GM analysis occurred simultaneously, potentially overlooking dynamic changes and their impact on metabolic health. Longitudinal studies tracking these changes would provide more robust insights and elucidate potential temporal relationships. Future studies employing larger sample sizes and longitudinal microbiome analyses are highly desirable. Third, the uncontrolled variation in breed, diet, age, BCS, and environment experienced by client-owned dogs, and the potential impact of other medications, introduced potential confounding factors. Our beta diversity analysis aimed to minimize this impact. However, future studies should implement stricter controls, including using dogs of the same breed, as well as dietary and environmental controls to isolate the effects of HAC on GM. We restricted the participating dogs from consuming table food and solely provided prescription or commercial diets with consistent ingredients, minimizing variation in dietary intake to overcome this limitation.

In addition, systemic antibiotics and probiotic supplements, which affect GM diversity and composition, were restricted throughout the study to isolate the effects of HAC on GM and reduce potential confounding influences from medication use. This study establishes a correlation between HAC and GM dysbiosis. However, causal relationships between dysbiosis and HAC metabolic abnormalities remain unconfirmed and require further investigation.

## 5. Conclusions

This study identifies notable differences in GM composition in dogs with HAC compared to healthy controls. The alterations in bacterial taxa, including the overrepresentation of Proteobacteria Actinobacteria, *Bacteroides*, *Enterococcus*, *Corynebacterium*, *Escherichia*, and *Proteus*, alongside an underrepresentation of Firmicutes, suggest a potential link between HAC and GM composition. This study’s findings emphasize the need for further research to clarify the associations between cortisol levels, gut dysbiosis, and metabolic abnormalities in HAC, which could potentially inform therapeutic interventions to manage metabolic abnormalities in affected dogs by targeting GM. This study offers valuable insights; however, larger sample sizes and longitudinal analyses are necessary to validate these findings and establish causality between specific bacterial communities and metabolic abnormalities in dogs with HAC.

## Figures and Tables

**Figure 1 animals-14-02883-f001:**
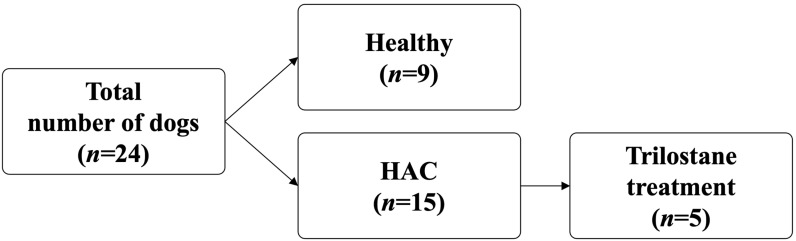
Study design.

**Figure 2 animals-14-02883-f002:**
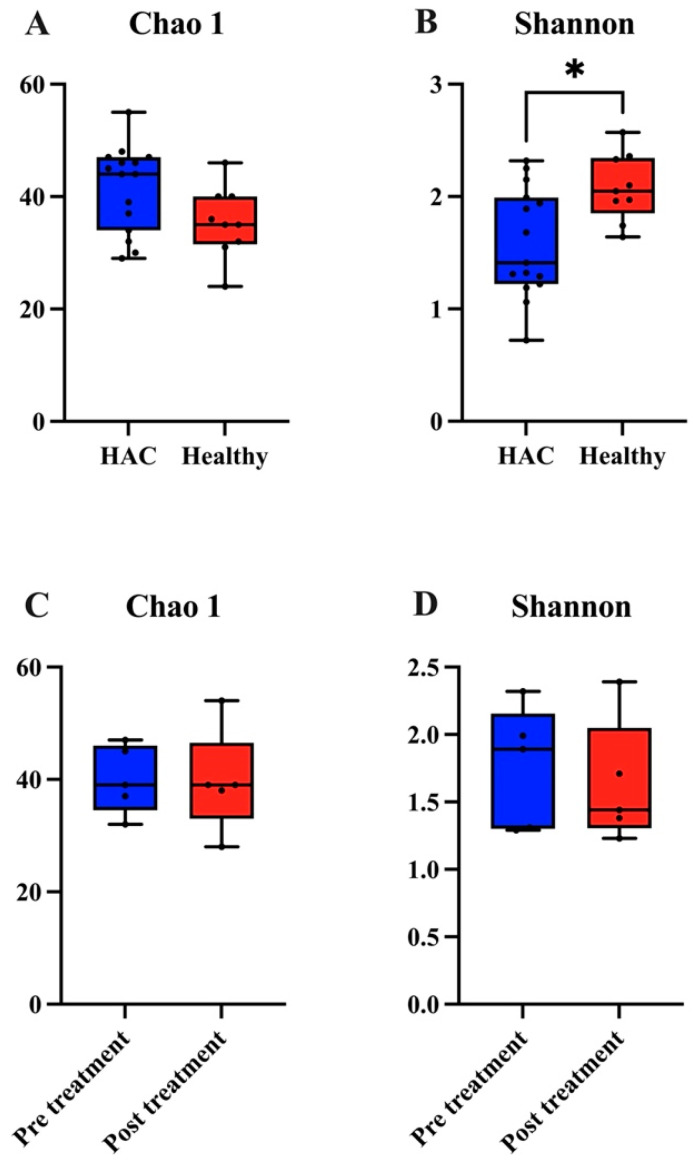
Gut microbiome alpha diversity analysis. Boxplot illustrating the differences in gut microbial alpha diversity. (**A**) The Chao1 index, a richness estimator focused on the total number of species, showed no significant difference between healthy dogs and dogs with HAC (Mann–Whitney test, *p* = 0.086). (**B**) In contrast, the Shannon index, which accounts for species richness and evenness, revealed significantly lower diversity in dogs with HAC (Mann–Whitney test, *p* = 0.0148 *). Boxplots depicting gut microbial alpha diversity before and after trilostane treatment, as measured by (**C**) the Chao 1 index and (**D**) the Shannon index, showed no significant differences (Wilcoxon signed-rank test, *p* > 0.05). Boxes represent the 25th–75th percentile of the distribution, the median is indicated by a thick line in the middle of the box, whiskers extend to 1.5 times the interquartile range, and individual data points are represented as dots. P-values are provided below each graph. HAC, hypercortisolism. * *p* < 0.05.

**Figure 3 animals-14-02883-f003:**
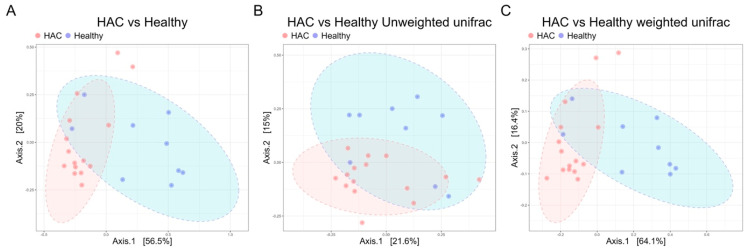
Distinct gut microbiome composition in dogs with HAC. Principal coordinate analysis plots demonstrate significant differences in microbial community composition between healthy dogs (*n* = 9) and dogs with HAC (*n* = 15) across all three distance metrics: (**A**) Bray-Curtis, (**B**) unweighted UniFrac, and (**C**) weighted UniFrac dissimilarity at the genus level. HAC, hypercortisolism.

**Figure 4 animals-14-02883-f004:**
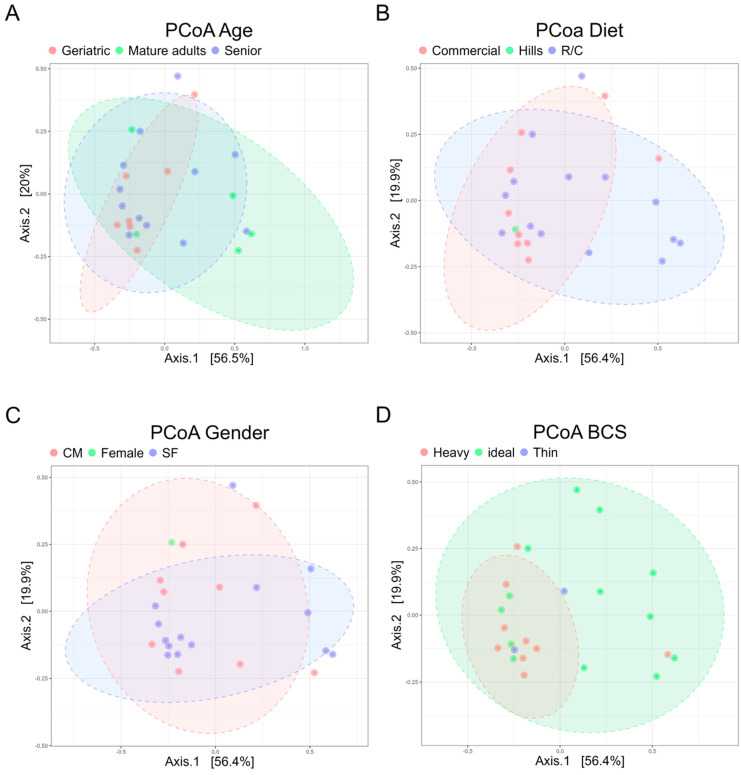
There were no significant differences in gut microbiome according to age, diet, sex, or body condition score across all dogs. Principal coordinate analysis with Bray-Curtis distance metrics of the microbiome in dogs across various categories-(**A**) age, (**B**) diet, (**C**) sex, and (**D**) BCS-revealed no significant differences. R/C, Royal Canin; CM, castrated male; IF, intact female; SF, spayed female; BCS, body condition score.

**Figure 5 animals-14-02883-f005:**
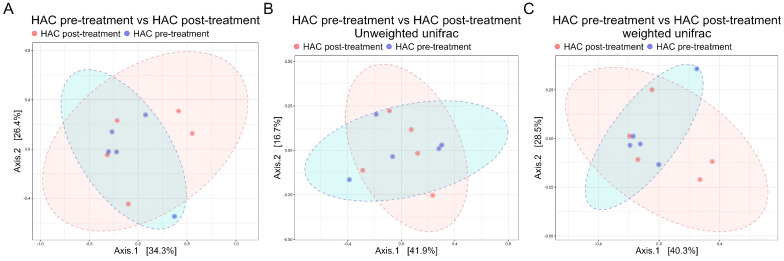
Gut microbiome composition remains stable in dogs with HAC after trilostane treatment. Principal coordinate analysis plots show no significant differences in microbial community composition of dogs with HAC pre- and post treatment (*n* = 5) across all three-distance metrics: (**A**) Bray-Curtis, (**B**) unweighted UniFrac, and (**C**) weighted UniFrac dissimilarity at the genus level (PERMANOVA, *p* > 0.05). HAC, hypercortisolism.

**Figure 6 animals-14-02883-f006:**
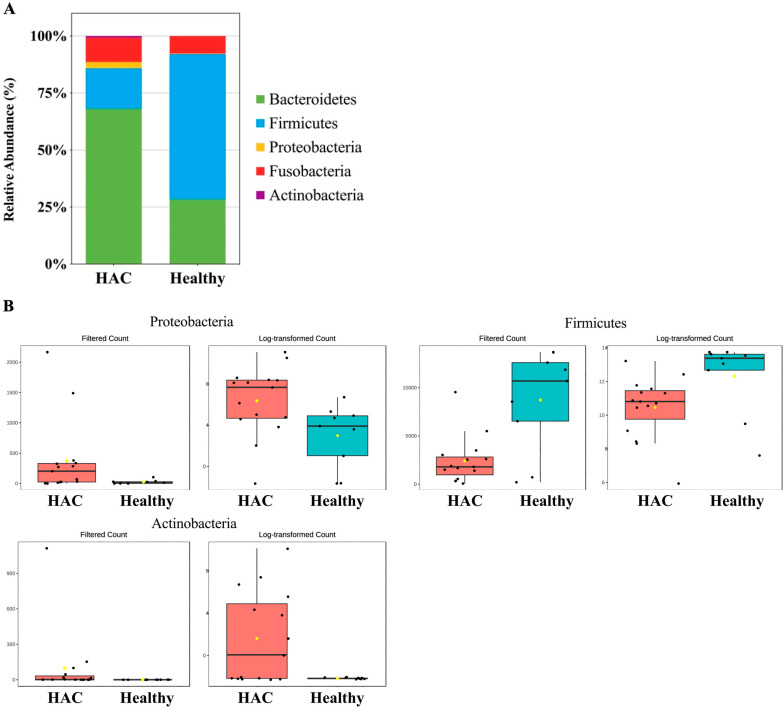
Comparison of microbiome taxa in healthy dogs (*n* = 9) and dogs with HAC (*n* = 15) at the phylum level. (**A**) A stacked bar graph showing the mean relative abundance of the gut microbiome at the phylum level. (**B**) Dogs with HAC have a significantly higher relative abundance of Proteobacteria and significantly lower levels of Firmicutes, with a significant increase in Actinobacteria compared to healthy controls. HAC, hypercortisolism.

**Figure 7 animals-14-02883-f007:**
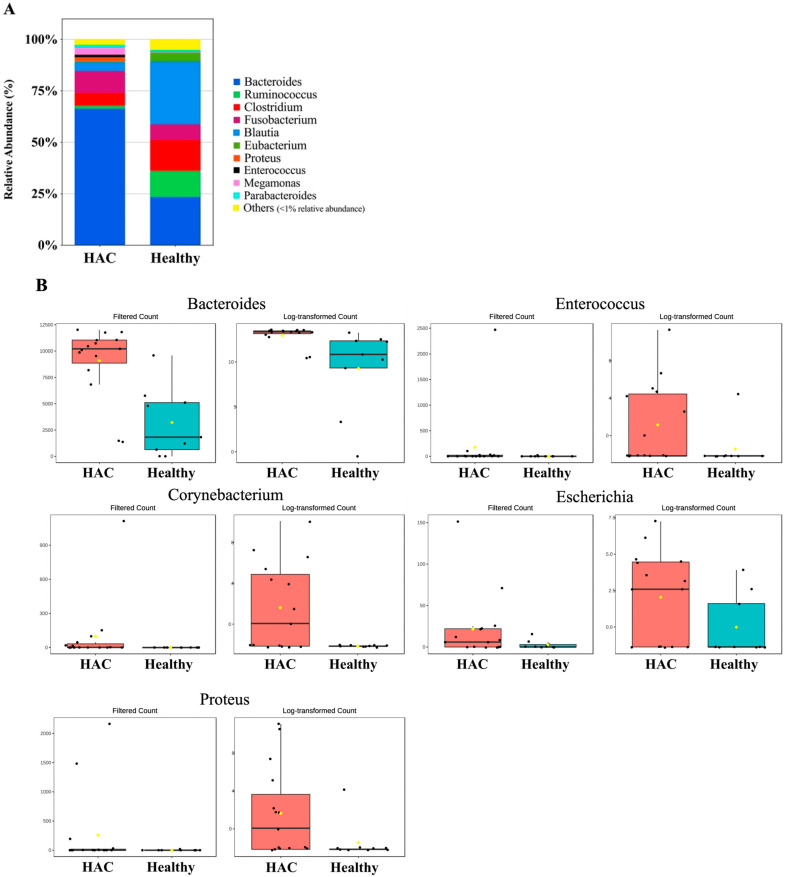
Comparison of microbiome taxa in healthy dogs (*n* = 9) and dogs with HAC (*n* = 15) at the genus level. (**A**) A stacked bar graph showing the mean relative abundance of the gut microbiome at the genus level. (**B**) Dogs with HAC show significantly higher levels of *Bacteroides*, *Enterococcus*, *Corynebacterium*, *Escherichia*, and *Proteus* compared to healthy controls. HAC, hypercortisolism.

**Figure 8 animals-14-02883-f008:**
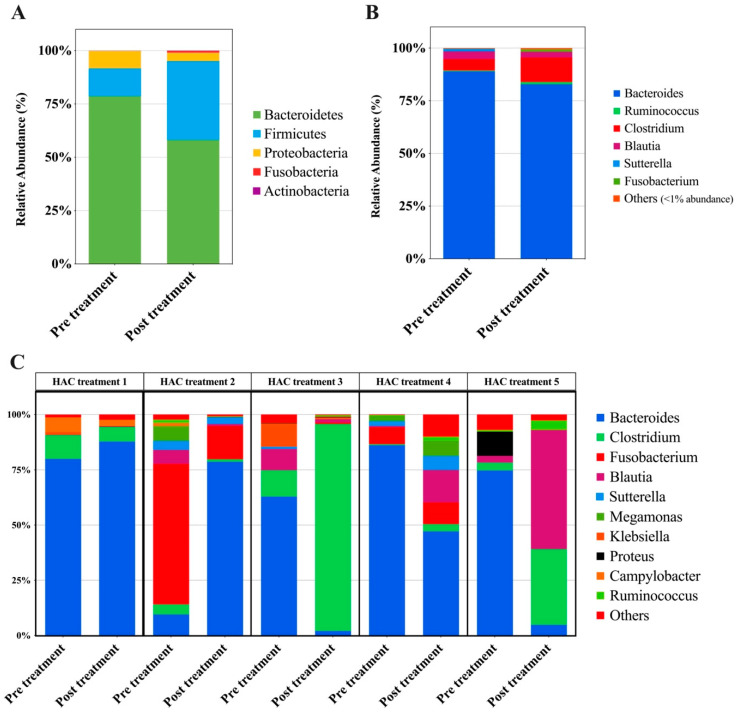
Analysis of bacterial communities in dogs with HAC before and after trilostane treatment. A stacked bar graph showing the mean relative abundance of the gut microbiome. At the phylum (**A**) and genus (**B**) levels, no significant changes were observed (DESeq2, *p* > 0.05). (**C**) Stacked bar plot of the 10 most abundant genera in HAC treatment dogs (*n* = 5). The mean relative abundance profiles of individual microbiomes before and after treatment are shown in sets, with no discernible trend between pre-treatment (left) and post-treatment (right) compositions. HAC, hypercortisolism.

**Table 1 animals-14-02883-t001:** Questionnaire for the companions of dogs with HAC to assess therapy satisfaction and report clinical signs [16].

**Please indicate how these statements currently apply to your dog**	**Not at all** **present** **1**	**A little** **present** **2**	**Somewhat** **present** **3**	**Very** **present** **4**	**Extremely** **present** **5**
My dog drinks too much	◻	◻	◻	◻	◻
My dog urinates frequently and urinates large amounts of urine	◻	◻	◻	◻	◻
My dog is always hungry	◻	◻	◻	◻	◻
My dog is panting a lot	◻	◻	◻	◻	◻
**Please indicate how these statements describe your assessment of the treatment with trilostane in your dog**	**Not at all** **present** **5**	**A little** **present** **4**	**Somewhat** **present** **3**	**Very** **present** **2**	**Extremely** **present** **1**
I am satisfied with the treatment of my dog	◻	◻	◻	◻	◻

**Table 2 animals-14-02883-t002:** Demographics, complete blood cell count, serum, and urine analysis in healthy dogs and dogs with HAC.

Variables (Reference Interval)		Healthy Dogs (*n* = 9)	HAC Dogs (*n* = 15)	*p*-Value
Age (years)		7.00 (6.00, 9.50)	11.00 (9.00, 13.00)	0.048 *
Sex	CM	4	5	
	IF	0	1	
	SF	5	9	
Breeds	Bichon Frise	1	1	
	Maltese	1	2	
	Miniature Schnauzer	0	1	
	Mix	1	3	
	Pomeranian	2	3	
	Poodle	4	2	
	Shihtzu	0	1	
	Yorkshire terrier	0	2	
Fecal score (1–7)		2.00 (2.00, 2.00)	2.00 (2.00, 3.00)	0.382
BCS (1–9)		5.00 (5.00, 5.00)	6.00 (5.00, 7.00)	0.114
BW (kg)		6.00 (3.50, 8.08)	5.12 (3.50, 7.32)	0.872
Systolic blood pressure (mmHg)		130.0 (120.0, 145.0)	150.0 (140.0, 165.0)	0.020 *
Baseline cortisol (0.5–10 ug/dL)			5.35 ± 3.20	
Post-ACTH cortisol (6–18 ug/dL)			27.40 (25.20, 30.00)	
PCV (37.3–61.7%)		50.20 (45.35, 56.50)	51.80 (41.80, 53.60)	0.861
PLT (148–484 K/uL)		266.0 (193.0, 358.5)	414.0 (307.0, 603.0)	0.025 *
Serum ALP (23–212 U/L)		48.0 (34.0, 110.5)	324.0 (96.0, 432.0)	0.002 ***
Serum ALT (10–125 U/L)		49.0 (30.5, 76.5)	97.0 (63.0, 147.0)	0.015 *
Serum CHOL (110–320 mg/dL)		184.5 (155.5, 227.5) [8]	217.0 (182.0, 269.0)	0.154
Serum TG (10–100 mg/dL)		83.0 (65.0, 131.0) [5]	102.0 (92.5, 131.0) [14]	0.289
Serum glucose (74–143 mg/dL)		99.0 (87.5, 105.0)	113.0 (107.0, 119.0)	0.006 **
USG (1.015–1.050)		1.030 (1.018, 1.045) [4]	1.030 (1.018, 1.045) [13]	0.491

Variables are expressed as mean ± SD or median (interquartile range) and [sample number]. HAC, hypercortisolism; CM, castrated male; IF, intact female; SF spayed female; BCS, body condition score; BW, body weight; PCV, packed cell volume; PLT, platelet; ALP, alkaline phosphatase; ALT, alanine aminotransferase; CHOL, cholesterol; TG, triglyceride; USG, urine specific gravity. * *p* < 0.05; ** *p* < 0.01; *** *p* < 0.005, statistically significant differences based on Mann–Whitney U test.

**Table 3 animals-14-02883-t003:** Demographics, complete blood cell count, serum, and urine analysis before and after trilostane treatment in dogs with HAC.

Variables (Reference Interval)		HAC Dogs (*n* = 5)		*p*-Value
		Pre-Treatment	Post-Treatment	
Age (years)		11.0 ± 2.8		
Gender	SF	5		
Breeds	Bichon Frise	1		
	Maltese	1		
	Mix	1		
	Pomeranian	1		
	Poodle	1		
Trilostane maintenance dosage (mg/kg)		1.45 (1.00, 2.00)		
Trilostane therapy duration (days)		41.0 (21.0, 67.5)		
Fecal score (1–7)		2.0 (2.0, 3.0)	2.0 (2.0, 2.5)	0.317
BCS (1–9)		5.0 (5.0, 5.5)	5.0 (5.0, 5.5)	1.000
Clinical signs score (5–25)		17.2 ± 3.0	9.4 ± 3.3	0.035 *
BW (kg)		4.78 ± 1.48	4.53 ± 1.52	0.137
Baseline cortisol levels (0.5–10 ug/dL)		3.12 ± 1.87	2.40 ± 1.37	0.594
Post-ACTH cortisol levels (6–18 ug/dL)		27.9 ± 2.1	6.32 ± 1.57	<0.0001 ***
Systolic blood pressure (mmHg)		148.0 ± 10.4	128.8 ± 16.4	0.007 **
PCV (37.3–61.7%)		48.9 ± 5.2	47.3 ± 3.0	0.550
PLT (148–484 K/uL)		481.4 ± 157.3	402.2 ± 77.9	0.391
Serum ALP (23–212 U/L)		242.4 ± 116.7	224.6 ± 116.5	0.788
Serum ALT (10–125 U/L)		123.8 ± 72.4	74.8 ± 21.3	0.203
Serum CHOL (110–320 mg/dL)		240.2 ± 46.3	216.4 ± 60.8	0.566
Serum TG (10–100 mg/dL) [*n* = 3]		91.7 ± 10.5	121.0 ± 53.0	0.355
Serum glucose (74–143 mg/dL)		121.4 ± 35.5	111.4 ± 7.6	0.494
USG (1.015–1.050) [*n* = 3]		1.03 ± 0.02	1.04 ± 0.02	0.148

The Variables are expressed as mean ± SD or median (interquartile range). HAC, hypercortisolism; SF, spayed female; BCS, body condition score; BW, body weight; PCV, packed cell volume; PLT, platelet; ALP, alkaline phosphatase; ALT, alanine aminotransferase; CHOL, cholesterol; TG, triglyceride; USG, urine specific gravity. * *p* < 0.05; ** *p* < 0.01; *** *p* < 0.005, statistically significant differences based on paired *t*-test and Wilcoxon signed-rank test.

## Data Availability

The data supporting the findings of this study are available from the corresponding author upon reasonable request.

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
