# Peer review of "Altered Gut Microbiome Composition in Dogs with Hyperadrenocorticism: Key Bacterial Genera Analysis"

_animals, 2024, doi:10.3390/ani14192883_

Round 1

Reviewer 1 Report

Comments and Suggestions for Authors

The paper showed the microbiota changes of dogs with hyperadrenocorticism and  healthy dogs, pre-treatment dogs and post-treatment dogs, which was interesting. 

1.     Title is not proper, for the microbiota was tested using fecal not gut content, please check.

2.     L15 Grammar mistake

3.     L41 please check the word “inn”

4.     L47  “successfully” should use the adj. type

5.     L82 Please add the statement of the dogs in the text including the age, variety, sex…, like L215-217

6.     L111-The method of how the blood is collected is not clearly stated, also the methods for testing the

7.     L87-88 regulation of ingredient composition, please make it more clear;

8.     Table 1 showed two tables, combined?

9.     L155 microbiome samples? not exact, please check; also, how the samples were stored should be added.

10. L198 check word “stringr, phangan…”

11. L202 please add the version of SPSS

12. L210, L218-220 “p” should italic; wrong word check

13. Table 2, please unify significant digit…also, the definition of “**” was incorrect, please check

14. Table 2, The data should be expressed as mean ± SD, please check  

15. Table 3, please unify significant digit…also, the definition of “**” was incorrect, please check, *** should also be added

16. L250 word “structure” is not proper here

17. L 258 grammar mistake, also, L258-260 should be moved to the discussion part.

18. L264 grammar mistake

19. L277 Figure not Figures

20. Figure B-D lack information, Please check

21. The definition of Figure 5 could be more clear, please modify

22. L326 HAC showed significant increase in…

23. L328 grammar mistake

24. L340, 349 no need for the definition of “**, ***”

25. L367 phylum level shouldn’t be italic, just genus level need

26. L457-465 The test didn’t refer to probiotic, so there is no need for much discussion of probiotics.

27. In the discussion part, the SCFA was also referred, so why not tested SCFAs in the test. If have, please add. If not, the related discussion could be deleted.

Comments on the Quality of English Language

There are many grammar mistakes in the manuscripts, and I recommend the paper polished and checked further.

Reviewer 2 Report

Comments and Suggestions for Authors

Dear Authors, I find your work really interesting and useful for the field. I have serious concerns however on the GM analysis topic. You'll find the details below. I really think you must address these concerns to improve the manuscript quality. My main concerns are : you must use a multivariate statistical analysis design for your betadiversity analysis. You must use a multi-test correction when using a differential abundance analysis. So either you did it and you must describe the procedures in your methodology, or you did not and this unfortunately make your results dubious.

Lines 28-31 : as a general rules, your use of the species term is confusing. Proteobacteria was a phylum name and Bacteroides is a genus name. So do you speak about specific species inside these groups, or do you employ the species term as a generic term. Except if you have specific species populations in mind, it would be better to substitue "species" by "population" or "group".

Line 163 : I never store fecal sample at 4°C without using a specific buffer for DNA stabilization.  If you don't have such buffer in your collection tube, storing fecal material <20°c is mandatory because otherwise storage condition has an impact on the GM composition.  In your study, it is not clear how you manage this storage phase. At the very least, you should include the period of storage before DNA extraction.

Line 173 : please, add the sequences for your V3V4 primer design.

Line 187 : you should add details about amplicon size, the quality rules you used for read filtering...

Line 191. I'm quite concerned about the use of Greengenes v13.5 and MicroSEQ database. These databases are clearly outdated (2013 !) Even Qiime2 protocol advise to use greengenes2 instead of Greengenes. This is a big issue if you want to compare your results with other publication whose GM analysis is based upon newer database.
    - The consequences of your choice of taxonomy database is that the names of your phylum are not up to date : Proteobacteria is now Pseudomonadota, Firmicutes is now Baccillota, Bacteroidetes is now Bacteroidota...
    - for an easy check : see http://lpsn.dsmz.de

Line 197-198, you list several R packages. could you please specify the purpose of each package in your work ?

Line 234 : why did you treat only 5 HAC dogs out of the 15 ?

Lines 250-253 :your description of gm analysis is really too short. what's the coverage ? how many reads did you analyse for each sample ? this part is important to support your alpha and beta analyse. Otherwise, I just don't believe your results... Alternatively, you can add more details in the methods part.

Line 255-257 : Small comment, I would specify that Chao1 is a richness estimator. Moreover, I fail to see any evidence for evenness analysis... Thus you can't say that evenness is lower without any further explanation.
Line 255 and Figure 2 : Shannon is an estimator of alpha diversity (actually an entropy calculation), Chao1 is an estimator of the unseen part of the species richness. So you cannot compare both regarding biodiversity because Shannon take into account the evenness, but Chao1 don't.  Moreover, Your description regarding the sensitivity a rare population for Chao1 but not Shannon is false. The rarity of a population in a sample, is inversely proportional to its impact in Shannon calculation. thus rare population have more weight comparatively. Please change your statements accordingly.

Line 280 : Details regarding beta diversity analysis are lacking in the methods. I understand using Bray-Curtis and unweighted Unifrac dissimilarity matrices, but Weighted unifrac is just redundant to Bray-Curtis.
    - why did you choose to use them all ?
    - Please add the details regarding permanova in the methods part.
    - Why did you skip the betadispersion analysis ?

Line 297 : Ok, did you actually test your variables (sex, diet, age) for clustering separately ? Did you use a multivariate analysis ?
    !! If you don't, this is unacceptable because you actually don't account for linked effect of variables in your model.
    -> There are multiple alternative multivariate analysis available in vegan for you.
    -> I would also test a dbrda model as a constraint ordination model to account for your multiple variables.

Lines 300 and following. PcoA is an ordination model, it does show your samples, but this is not in any case a statistical test. Your sentences are confusing, permanova show differences or not and pcoa can only illustrate or suggest !
    - Please, correct your sentences accordingly.
Line 334 and the corresponding part in the methodology.
    - What is the multitest correction procedure you used for your differential abundance analysis (DAA) ?  They are available in Prism but there is no detail about it ?
    -> If you did not use such correction procedure, I just don't believe your results! thus you must have a multitest correction.
    -> As an alternative I'm using DESeq2 in R for DAA. there are also alternative libraries accounting for paired samples in R. All of them includes false discovery rate correction procedures.

Reviewer 3 Report

Comments and Suggestions for Authors

This study identifies alterations in the dog gut microbiome in those affected with Hyperadrenocorticism (HAC) in contrast to healthy controls. 

Specifically, there is an observed proliferation of bacteria belonging to the Proteobacteria and Bacteroides, alongside a reduced presence of Firmicutes, 

Ruminococcus, and Blautia in dogs with HAC. This suggest a potential correlation between HAC and changes in gut microbiota composition.

Rephrase: with more harmful bacteria and fewer beneficial ones.

What was the steroid use history in the healthy group? 

I think this paper is a simple study but the conclusions have to be taken very carefully, As can be seen from table, Figure 7 c, none of the pre treatment dogs were the same. While this is only 5 dogs, one does have to take in the argument if concurrent GI disease was present and not just HAC.  Breeds listed know for GI or other system disease include Miniature Schnauzers, Yorkshire terriers etc. so a cobalamin and folate level would have been useful along with the diarrhea or even ultrasound changes to make sure there was no IBD/CE or chronic pancreatitis. 

Reviewer 4 Report

Comments and Suggestions for Authors

Overall, this study is well-designed, but it needs to be improved in light of the following comments. 

Major comments.

1. Gut microbiota can be very specific in different breeds of dogs. Why the dogs from the same breed were not included in this study? This is a limitation and must be incorporated into the discussion section. 

2. What can be the effect of different seasons and seasonal diet/temp? variations on the microbiota. Since the dog's enrollment was spanned for a year, did authors analyze any variation in samples collected in different seasons or why were the samples not collected from a shorter span.

3. Brief results on abundance at levels other than phylum and genus should be added to the description of the results. 

4. Discussion needs to be strengthened by adding studies from dogs. 

5. Conclusion needs to be toned down

Minor comments. 

Line 22. Define GM in full first.

Line 24. Remove Figure 1 from the abstract.

Line 28. Remove PERMANOVA

Line 29. Refer to the correct taxonomic level for Proteobacteria and Bacteroides. Could you tell me if you're referring to species?

Line 31. Clarify (persistence) at what day post-treatment?

Line 32: Did you find a difference only at the genus level? If not, why to focus here? Did you measure the numerical diversity of the important genera using real-time PCR?

Line 53. Add a few more references

Line 66. is limited.  Add references to those limited studies and also give those findings

67. Describe the duration of this persistence. 

68. Remove god owner and use companion/caregiver/custodian or any other appropriate word.

Line 157. Rewrite the sentence to clarify that you collected samples only for one time after treatment. How long after treatment were the samples taken for microbiome study? 

Line 326. Replace "increase in" with a higher

line 361. Add in animals studied specifically. 

Line 368, Add if there is any study from animals especially dogs

Comments on the Quality of English Language

Minor corrections are needed

Round 2

Reviewer 1 Report

Comments and Suggestions for Authors

1. The layout of Fig. 6 and Fig.7 is not proper.

2. Please add the information to the acknowledgment. If no, please complete.

Reviewer 2 Report

Comments and Suggestions for Authors

Dear authors, I thank you for your careful editing of the manuscript. the methods description is clear and the tools used are correct for microbiota analysis.

My only is that, it would be nice to add a reference for QIIme and the taxonomic database you used.

with this last addition the manuscript is ok for publication, for me at least.